# Deciphering novel common gene signatures for rheumatoid arthritis and systemic lupus erythematosus by integrative analysis of transcriptomic profiles

**Neetu Tyagi[1,2], Kusum Mehla[1], Dinesh Gupta[1]** *

**1** Translational Bioinformatics Group, International Centre for Genetic Engineering and Biotechnology (ICGEB), New Delhi, India, **2** Regional Centre for Biotechnology, Faridabad, India

* dinesh@icgeb.res.in, dinesh.bioinfo@gmail.com

**Data Availability Statement:** All relevant data are within the paper and its Supporting Information files. In this study, we have used publicly available

## Abstract

Rheumatoid Arthritis (RA) and Systemic Lupus Erythematosus (SLE) are the two highly prevalent debilitating and sometimes life-threatening systemic inflammatory autoimmune diseases. The etiology and pathogenesis of RA and SLE are interconnected in several ways, with limited knowledge about the underlying molecular mechanisms. With the motivation to better understand shared biological mechanisms and determine novel therapeutic targets, we explored common molecular disease signatures by performing a meta-analysis of publicly available microarray gene expression datasets of RA and SLE. We performed an integrated, multi-cohort analysis of 1088 transcriptomic profiles from 14 independent studies to identify common gene signatures. We identified sixty-two genes common among RA and SLE, out of which fifty-nine genes (21 upregulated and 38 downregulated) had similar expression profiles in the diseases. However, antagonistic expression profiles were observed for ACVR2A, FAM135A, and MAPRE1 genes. Thirty genes common between RA and SLE were proposed as robust gene signatures, with persistent expression in all the studies and cell types. These gene signatures were found to be involved in innate as well as adaptive immune responses, bone development and growth. In conclusion, our analysis of multicohort and multiple microarray datasets would provide the basis for understanding the common mechanisms of pathogenesis and exploring these gene signatures for their diagnostic and therapeutic potential.

## Introduction

Autoimmune diseases are a family of more than 80 chronic, often debilitating, and sometimes life-threatening illnesses; some of which are well characterized such as Rheumatoid Arthritis (RA), Systemic Lupus Erythematosus (SLE), type 1 diabetes, multiple sclerosis, and psoriatic arthritis while some are rare and difficult to diagnose [1]. Epidemiological data provide evidence of a steady increase in autoimmune diseases globally, from an estimated prevalence of

data from the NCBI Gene Expression Omnibus repository (accession numbers GSE15573, GSE4588, GSE17755, GSE1402, GSE56649, GSE93272, GSE68689, GSE11909, GSE50772, GSE22098, GSE61635, GSE24060).

**Funding:** This work was supported by the bioinformatics infrastructure grant from the Department of Biotechnology, Government of India (no. BT/PR40151/BTIS/137/5/2021).

**Competing interests:** On behalf of all authors, the corresponding author states that there is no conflict of interest.

3.2% between 1965 and 1995 to 19.1 ± 43.1 reported in 2018 [2, 3]. In a recent study, the risk of COVID-19 in patients with autoimmune diseases was reported to be significantly higher than in control patients [4].

RA is a multisystem chronic inflammatory disease characterized by erosive synovitis, autoantibody production (rheumatoid factor, RF), polyarticular inflammation of small joints of the hands, wrist, and feet, and associated stiffness and organ damage, leading to severe complications and poor quality life [5]. SLE is another chronic autoimmune disease with various clinical manifestations that affect multiple organs and tissues and involves a complex interaction between various immunological, environmental, hormonal, and genetic factors [6, 7]. Prior clinical and epidemiological studies provided evidence that both RA and SLE have overlapping clinical symptoms and shared genetic architecture [8]. They share certain clinical and pathogenic features, including activation of B and T cells, immune cell (macrophages and neutrophils) migration and infiltration of organs, production of a variety of pathogenic autoantibodies/inflammatory cytokines, and several susceptibility loci [9, 10]. The treatments are very similar for autoimmune disorders, except for cases involving organ damage or where the features of one disease dominate over the other. Thus, elucidating these shared genetic determinants would eventually contribute to identifying biomarkers [11, 12] and developing novel therapeutic strategies for combined diagnosis and prognosis of RA and SLE. The gene expression patterns analysis can provide valuable details for better understanding of molecular mechanisms in the diseases. To gain more rational and decisive results related to different autoimmune diseases, several studies have previously focused on analyzing integrated data from various studies for a single disease [6, 13–17]. Additionally, meta-analysis techniques offer tremendous opportunities to integrate data from different diseases to reveal novel common gene signatures, which may be missed in single disease meta-analysis studies. In the context of RA and SLE, Tuller et al. [18] analyzed the publicly available data from the PBMC samples of six different autoimmune diseases (SLE, multiple sclerosis, RA, juvenile RA, type 1 diabetes, Crohn's disease and ulcerative colitis). The study aimed to understand the intra-regulatory mechanism in PBMC, which can be common to all autoimmune diseases or specific to any few of them. They found certain chemokines and interleukin genes were differentially expressed in the analyzed autoimmune diseases. Silva et al. [19] integrated the SLE and RA expression datasets and profiling modules for specifically induced or repressed and comodulated genes to uncover the coexpression patterns. Higgs et al. [20] conducted a study to analyze common signatures related to type 1 IFN by integrating data from SLE, myositis, RA and scleroderma. Toro-Domínguez et al. [21] uncovered the common signatures from SLE, RA and SjS (Sjogren's Syndrome) PBMC patients. They conducted the gene expression meta-analysis using the publicly available gene expression datasets. Wang et al. [22] identified eight differentially expressed genes associated with many rheumatic diseases, including RA, SLE, ankylosing spondylitis, and osteoarthritis. Luan et al. [23] conducted a study integrating microRNA, methylation, and expression datasets to study the shared and specific mechanisms of four autoimmune diseases. In this study, they discovered shared and disease-specific pathways. A recent study by Wang et al. [24] identified the dysregulation of megakaryocyte expansion contributing to the pathogenesis of many autoimmune diseases, including RA and SLE. However, no study to date has focused on systematically identifying the common factors and their role in the underlying mechanism of the two most common systemic autoimmune diseases, RA and SLE. Therefore, the aim of this study is to reveal the commonly dysregulated genes and the significant gene networks associated with the two frequent chronic rheumatic autoimmune diseases.

In this study, we analyzed 1088 publicly available microarray samples of the two diseases belonging to different cell types, ages, sexes, platforms, and genetic backgrounds. To identify

the common or specific gene expression signatures in these two diseases, we analyzed the large-scale multi-cohort gene expression microarray datasets of Peripheral Blood Mononuclear Cells (PBMCs), Whole Blood (WB), and other cell type samples obtained from SLE and RA patients. To our knowledge, this is the first large-scale study to report the meta-analysis of gene expression microarray datasets considering the biological and technical heterogeneity observed in the real-world patient population for the two systemic inflammatory diseases. We analyzed the common gene expression patterns, hub genes, commonly regulated important pathways, and regulatory biomarkers involved in the disease mechanism of RA and SLE.

## Methods

### Data collection

The microarray gene expression data was downloaded from the Gene Expression Omnibus (GEO) database (https://www.ncbi.nlm.nih.gov/geo/). The search terms used for the data retrieval include "rheumatoid arthritis" or "RA" and "systemic lupus erythematosus" or "SLE", each with the filters: organism (*Homo sapiens)*, study type (expression profiling by array) and entry type (dataset/series). As a result of the search, 538 datasets were retrieved. The retrieved datasets were filtered based on the presence of drug-treated samples, missing healthy controls, tissue type, unrelated/duplicated datasets, and summary Area Under the Receiver Operating Characteristic (AUROC) score to obtain a large and independent cohort of 1088 samples of RA and SLE patients. The dataset inclusion/exclusion criteria and detailed workflow of the study are shown in **Fig 1**. The selected datasets were downloaded from the GEO database using the GEOquery R package [25].

### Data preprocessing and meta-analysis

For meta-analysis, downloaded gene expression datasets from various studies were preprocessed using the quantile normalization method and imported into MetaIntegrator framework [26]. The MetaIntegrator-aided meta-analysis combines significance (P) values, Z-scores, ranks, or Effect Size (ES) across different studies and generates formal overall P values for each studied effect. It computes the Hedges g effect size for each gene in each dataset and pools these effect sizes across datasets from different studies.

$$\text{Hedges g effectsize (g)} = J \frac{\bar{X}_1 - \bar{X}_0}{\sqrt{\frac{(n_1-1)S_1^2+(n_0-1)S_0^2}{n_1+n_0-2}}}$$

Where:

J is the Hedges g correction factor; $\bar{X}_1$ and $\bar{X}_0$ are the mean expression values; $S_1$ and $S_0$ are the standard deviations; and $n_1$ and $n_0$ are the numbers of samples for case and control, respectively. The summary effect size $g_s$ was calculated using a random effect model using the equation given below.

$$\text{Pooled effect size} \left( g_s \right) = \frac{\sum_i^n W_i g_i}{\sum_i^n W_i}$$

n is the no. of studies, $g_i$ is the hedges' g of the gene within dataset i, $W_i$ is the weight calculated by $1/(V_i + T2)$, $V_i$ is the variance of the gene within a given dataset i, and T2 is the inter-dataset variation estimated by DerSimonian-laired method. MetaIntegrator computes the effect size for each data set independently, thus grabbing heterogeneity and avoiding the limitations of batch effect corrections. The random effect model employed in the above equation would

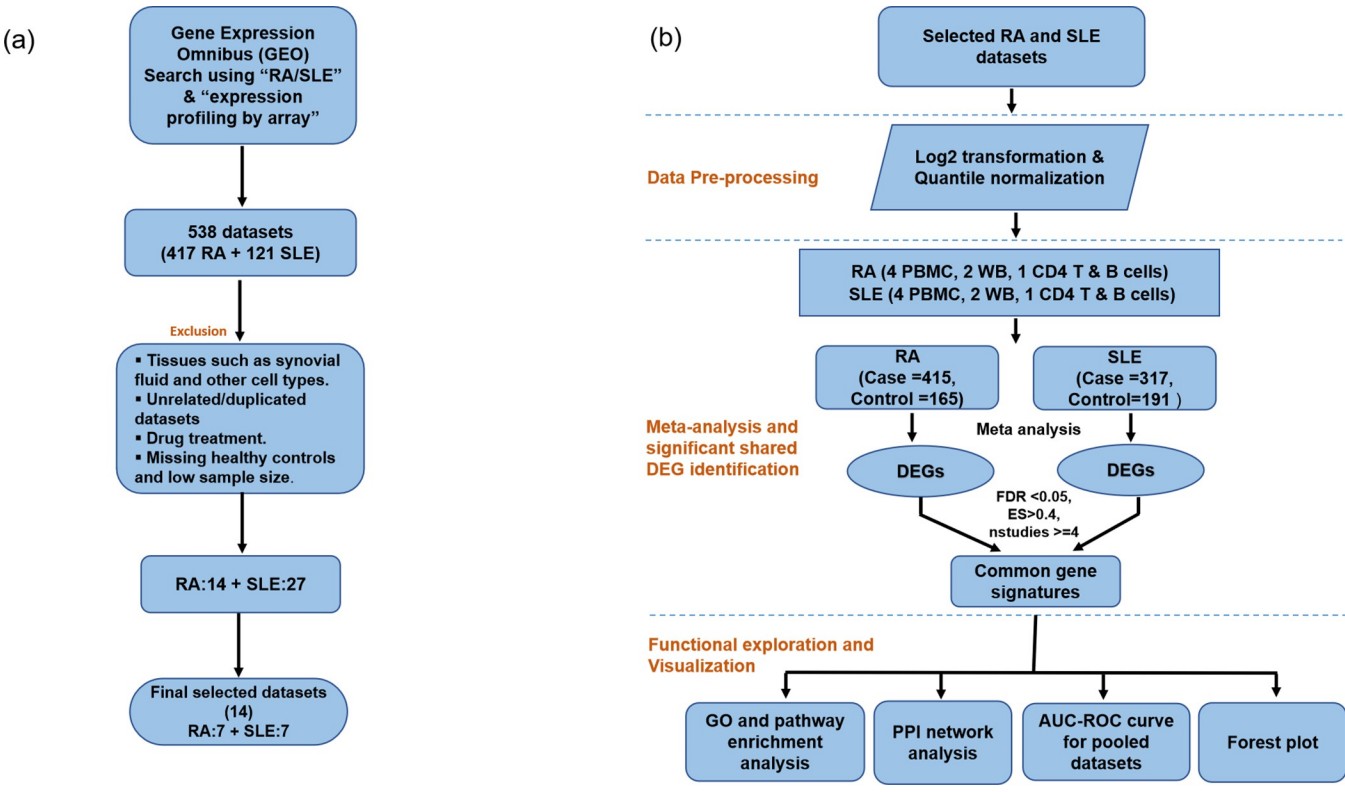

**Fig 1.** Meta-analysis workflow of the study: **(a)** Details of datasets collection. **(b)** Data preprocessing and meta-analysis.

provide more conservative results by extracting fewer Differentially Expressed Genes (DEGs) with more confidence. MetaIntegrator calculates Cochrane's Q value and a combined p-value using Fisher's method to account for the heterogeneity of ES estimates between the studies. In the second stage of filtering, we added/removed datasets one-by-one to optimize the AUROC score to generate a model of the pooled datasets with high summary/pooled AUROC scores while keeping in mind to balance the samples and datasets from different tissues. Detailed exclusion criteria followed in this study are given in S1 Table. In the process of data integration, patient samples from different sources were not segregated to reveal the common gene signature in these two autoimmune diseases. After the meta-analysis, a subset of the common DEGs was selected for downstream analysis using the filtering criteria: FDR <0.05, ES >0.40, and observed in at least four studies.

## Hub genes and network analysis

To generate the Protein-Protein Interaction (PPI) networks, NetworkAnalyst was used. The networks for the common gene signatures and the RA and SLE-specific top DEGs were generated using the reference innateDB interactome database [27]. The identified common gene signatures and top 50 DEGs from the independent meta-analyses of RA and SLE were used to construct their respective networks for identifying hub genes.

## Gene ontology and integrative pathway analysis

To identify over-represented biological terms and enriched pathways, we used the Enrichr R package [28]. The DEGs obtained from the independent meta-analyses of RA and SLE and the

common gene signatures revealed in our study were used as input for Enrichr. Default settings were used for the functional annotation and the p-value was calculated using Fisher's exact test. A significance threshold criterion of p-value <0.05 was used to identify significant gene ontology terms and biological pathways.

## Results

### Data preprocessing

The gene expression datasets, downloaded from the GEO, were manually checked to exclude duplicate and irrelevant studies. Out of the 538 studies, we selected the studies that reported gene expression in WB, PBMC, or blood cell components. We excluded studies. In the initial filtering step, total 83 studies were filtered out as the studies represented the effect of drug treatment in samples. From the remaining datasets, 29 studies were excluded as they lacked healthy controls. Further, we also removed studies involving other tissues, such as synovial fluid, chondrocytes or lung tissues. Finally, we were left with 38 datasets, representing 14 RA and 24 SLE studies. In the second filtering stage, the datasets that led to a decrease in summary AUROC were also excluded. After filtering, we were left with 14 definitive studies which included seven SLE datasets (GSE11909, GSE50772, GSE22098, GSE4588, GSE61635, GSE17755 and GSE24060) and seven RA datasets (GSE93272, GSE15573, GSE4588, GSE17755, GSE1402, GSE56649, and GSE68689). The resulting 14 datasets were biologically, clinically, and technically heterogeneous, representing five different countries, patients of different ages, different sample types (whole blood and PBMCs), and distinct technologies for gene expression profiling. A total of 1088 samples were used for identifying commonly dysregulated genes ideal for understanding the molecular pathogenesis of RA and SLE.

### Meta-analysis and identification of common gene signatures in RA and SLE

We identified and downloaded the publicly available GEO gene expression datasets to achieve an extensive, unbiased study of the common signatures between RA and SLE. From the initially available public datasets, we selected 14 studies that passed the inclusion criteria (see methods). The chosen studies consisted of seven datasets for RA (4 PBMC; 2 WB; 1 CD4 T and B cells) and 7 for SLE (4 PBMC; 2 WB; and 1 CD4 T and B cells), which included 580 samples for RA (415 RA patients and 165 controls) and 508 for SLE (317 SLE patients and 191 controls) as shown in **Fig 1**. A detailed summary of the included datasets and samples is given in **Table 1**. From the meta-analysis, we identified 377 significant DEGs for RA (135 upregulated, 242 downregulated) and 1175 for SLE (566 upregulated, 609 downregulated) with the filtering criteria set to ES>0.4, FDR<0.05, number of studies (nstudies> = 4) and AUROC scores. The final selected dataset involving 14 studies given in **Table 1**, had more discriminatory power, as evidenced by high summary AUROC and was eventually used for predicting the DEGs involved in shared molecular mechanisms of the two diseases. The meta-scores distinguish patient samples from the healthy controls with an AUROC of 0.887 (95% confidence interval (CI): 0.70–1) and 0.927 (95% CI: 0.73–1) for RA and SLE, respectively (**Fig 2A and 2B**). Precision curves for the RA and SLE datasets are shown in **S1 Fig**.

We identified 62 genes common to both RA and SLE (see **S2 Table**). Fifty-nine genes (21 upregulated and 38 downregulated) out of the common had similar expression profiles in both the diseases. However, antagonistic expression profiles were observed for the remaining three genes (ACVR2A, FAM135A, and MAPRE1). List of 50 most significant up or downregulated genes for RA and SLE are provided in **S3 Table.** Of the common genes, 30 were defined as gene signatures for both RA and SLE, as their expression was reported across all the studies. The Venn diagram highlights the unique and common genes of RA and SLE (**Fig 3**). A

**Table 1. Summary of the included RA and SLE datasets and their samples.**

| SLE datasets | | | | | | | |
|---|---|---|---|---|---|---|---|
| S. No. | GSE_ids | Source | Case | Control | Used Samples | Total Samples | PMID | Organization/Centre/City |
| 1 | GSE11909 | PBMC | 63 | 12 | 75 | 175 | 18631455 | Baylor University, Texas, USA |
| 2 | GSE50772 | PBMC | 61 | 20 | 81 | 81 | 25861459 | ITGR Diagnostics Discovery, South San Francisco, USA |
| 3 | GSE22098 | Whole blood | 40 | 43 | 83 | 274 | 20725040 | Baylor University, Texas, USA |
| 4 | GSE4588 | CD4 T and B cells | 26 | 15 | 41 | 49 | NA | Université catholique de Louvain, Institut de Recherches Expérimentales et Cliniques, Brussels, Belgium |
| 5 | GSE61635 | Whole blood | 99 | 30 | 129 | 129 | NA | Eli Lilly and Company, USA |
| 6 | GSE17755 | PBMC | 22 | 53 | 75 | 244 | 21496236 | Wakayama Medical University, Ibaraki, Japan |
| 7 | GSE24060 | PBMC | 6 | 18 | 24 | 80 | 21521520 | The National Institute of Environmental Health Sciences, Durham, USA |

| RA datasets | | | | | | | |
|---|---|---|---|---|---|---|---|
| S. No. | GSE_ids | Source | Case | Control | Used Samples | Total Samples | PMID | Organization/Centre/City |
| 1 | GSE15573 | PBMC | 18 | 15 | 33 | 33 | 19710928 | CEA IG, Gene Expression Platform, Evry, France |
| 2 | GSE4588 | CD4 B and T cells | 15 | 19 | 34 | 49 | NA | Université catholique de Louvain, Institut de Recherches Expérimentales et Cliniques, Brussels, Belgium |
| 3 | GSE17755 | PBMC | 112 | 53 | 165 | 244 | 21496236 | Wakayama Medical University, Ibaraki, Japan |
| 4 | GSE1402 | PBMC | 15 | 15 | 30 | 57 | 15150433 | Cincinnati Childrens Hospital Medical Center, Cincinnati, USA |
| 5 | GSE56649 | Peripheral blood CD4$^+$ T cells | 13 | 9 | 22 | 22 | 25880754 | Peking University, People's Hospital, Beijing, China |
| 6 | GSE93272 | Whole blood | 232 | 43 | 275 | 275 | 30013029 | Takeda Pharmaceutical Company Limited, Fujisawa, Japan |
| 7 | GSE68689 | Whole blood | 10 | 11 | 21 | 21 | NA | Selventa, Cambridge, USA |

heatmap of the 62 common genes between RA and SLE is shown in Fig 4. Heatmaps for highly differentially expressed RA and SLE genes are shown in S2 Fig. The complete description of the common gene signatures persistent across all datasets is given in Table 2.

## Hub genes network analysis

We have generated three interaction networks as described in the methods section. The interaction network for common genes comprises 53 seeds with 907 connecting nodes and 1143

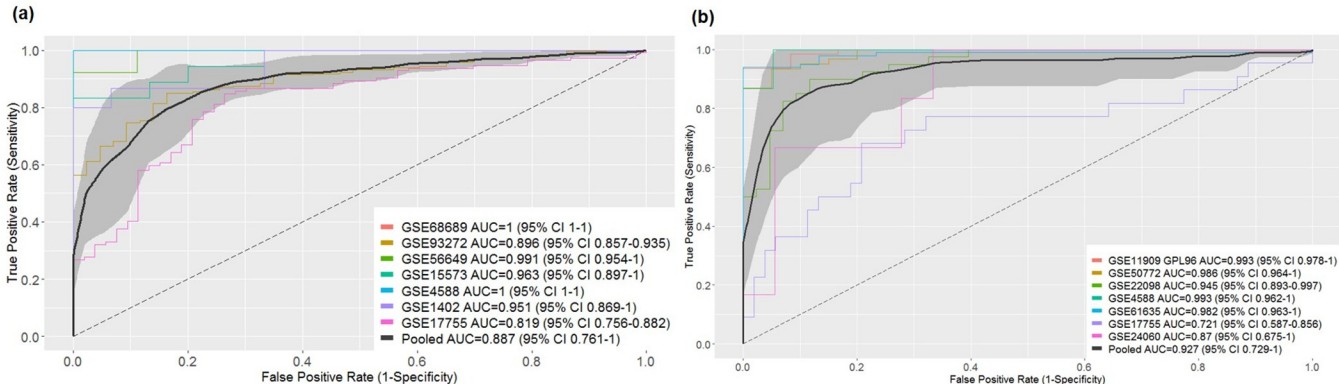

**Fig 2. Receiver operating characteristic curves of RA and SLE. (a)** RA datasets and **(b)** SLE datasets. A perfect classifier must have an AUROC of 1, while a random classifier has an AUC of 0.5. Here, the summary curve is a composite of the individual studies from PBMC, WB, and CD4 T and B Cells samples with AUROC scores of 0.887 and 0.927 for RA and SLE respectively.

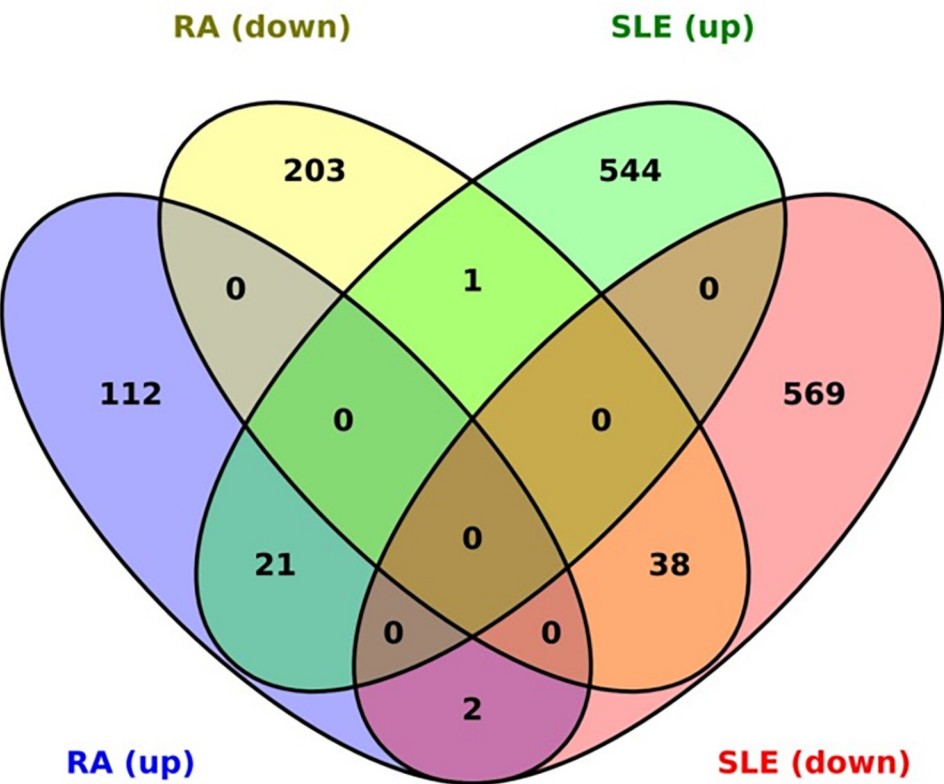

**Fig 3. Venn diagram of DEGs.** Comparison of DEGs (both upregulated and downregulated) obtained from individual meta-analyses of RA and SLE. The intersection showed the genes common to both diseases.

edges representing the interaction between these proteins. This analysis identified key hub genes among the common genes and the top DEGs specific to both RA and SLE.

The PPI network for the common gene signatures is shown in **Fig 5**. We analyzed the interaction network for the common gene signatures and found many hub genes based on the high degree of centrality and betweenness. Among these, the main hub genes were CDK1 (degree: 146, betweenness: 112412.8), RPS28 (degree: 94, betweenness: 66300.6), CCNA2 (degree: 77, betweenness: 46749.4), RBL2 (degree: 69, betweenness: 45832.2), EIF4B (degree: 54, betweenness: 44979.1) and MAPRE1 (degree: 42, betweenness: 34895.5).

From the interaction network of top 50 DEGs for RA, the hub genes with the highest centrality degree and betweenness were SMURF2 (degree: 88, betweenness: 48783.5), CCNA2 (degree: 77, betweenness: 41466), and B2M (degree: 75, betweenness: 46273) for the upregulated genes and EWSR1 (degree: 212, betweenness: 178696.9), MAPK3 (degree: 127, betweenness: 107148.4) and G3BP1 (degree: 93, betweenness: 68231.1) for downregulated genes.

For the SLE interaction network, among the top 50 DEGs the notable hub genes included STAT1 (degree: 223, betweenness: 163468.8), ISG15 (degree: 188, betweenness: 156960.3), and PLSCR1 (degree: 84, betweenness: 61314.9) for the upregulated genes and CBL (degree: 216, betweenness: 214523.2), STUB1 (degree: 169, betweenness: 171140.1), MAPK8 (degree: 140, betweenness: 152176.3) for the downregulated genes.

A detailed description of the hub genes for the common gene signatures and disease-specific genes (RA and SLE) is provided in **S4 Table**. The PPI networks of RA and SLE for the top 50 DEGs are shown in **S3** and **S4 Figs**. Forest plots were created for a few common genes (**Fig 6**) to represent the consistency of gene expression in both diseases across all datasets.

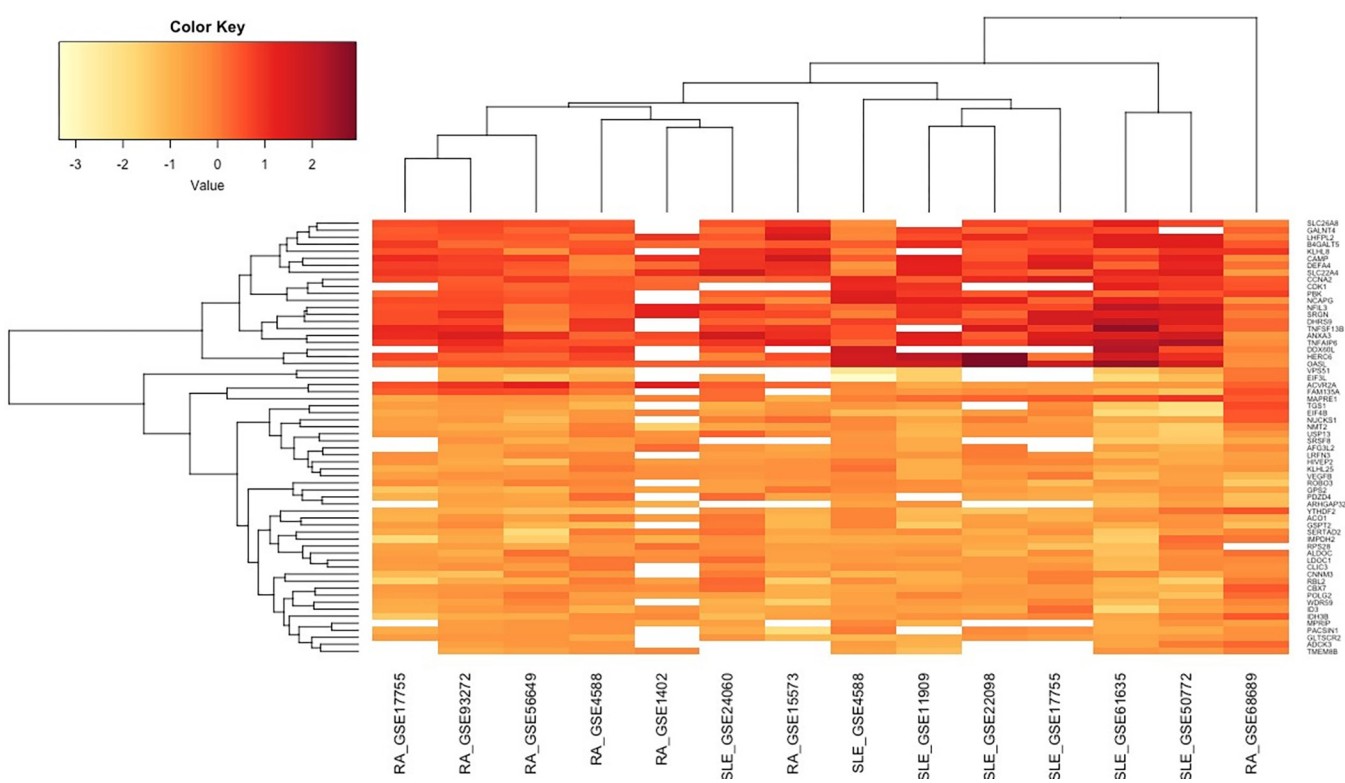

**Fig 4. Heatmap represent the effect size of the common differentially expressed gene across all datasets for RA and SLE.** Each column is a dataset and each row represents the expression level of the particular gene in all datasets. The colour scale represents the pooled effect size of that particular gene ranging from yellow (low expression) to red (high expression).

## Identification of over-represented biological pathways and gene ontology terms

The common gene signatures between RA and SLE were enriched with pathways related to TGF-beta signaling, viral carcinogenesis, citrate cycle, and cellular senescence. In RA, the NF-Kappa B signaling pathway, cytokine-cytokine receptor interaction, the IL-17 signaling pathway, and the rheumatoid arthritis pathways were observed for upregulated genes. In contrast, pathways such as the mTOR signaling pathway, the PI3K-Akt signaling pathway, and HIF-1 signaling pathways were observed for downregulated genes. In SLE, upregulated genes were enriched with pathways such as NOD-like receptor signaling, necroptosis, RIG-I-like receptor signaling, toll-like receptor signaling and many viral infections-related signaling pathways. In contrast, the downregulated genes were observed to show enrichment of pathways such as adipocytokine signaling, inflammatory mediator regulation of TRP channels, insulin-resistance and ubiquitin-mediated proteolysis. The top 10 Gene Ontology (GO) terms for each category *viz*. Molecular Function (MF), Cellular Component (CC), and Biological Process (BP) and enriched pathways for common genes are shown in **Fig 7**. Detailed information about significant GO terms, enriched pathways and genes involved for each category is provided in **S5 Table**.

In enrichment analysis for common genes, BP terms such as tricarboxylic acid metabolic process, regulation of translational initiation, innate immune response in mucosa, mitotic chromosome condensation mucosal immune response, and neutrophil degranulation were observed (see **Fig 7(A)**).

**Table 2. Detailed description of the common gene signatures persistent across both diseases, and cell types.**

| Gene | Gene name (detail) | Gene description | RA combined effect size | SLE combined effect size |
|------|-------------------|------------------|------------------------|-------------------------|
| ACO1 | aconitase 1 (ACO1) | The protein encoded by this gene is a bifunctional, cytosolic protein that functions as an essential enzyme in the TCA cycle and interacts with mRNA to control the levels of iron inside cells. The encoded protein has been identified as a moonlighting protein based on its ability to perform mechanistically distinct functions. | -0.590595045 | -0.50551105 |
| ACVR2A | activin A receptor type 2A (ACVR2A) | This gene encodes a receptor that mediates the functions of activins, which are members of the transforming growth factor-beta (TGF-beta) superfamily involved in diverse biological processes. The encoded protein is a transmembrane serine-threonine kinase receptor which mediates signaling by forming heterodimeric complexes with various combinations of type I and type II receptors and ligands in a cell-specific manner. | 0.709280042 | -0.435212793 |
| ALDOC | aldolase, fructose-bisphosphate C (ALDOC) | This gene encodes a member of the class I fructose-biphosphate aldolase gene family. Expressed specifically in the hippocampus and Purkinje cells of the brain, the encoded protein is a glycolytic enzyme that catalyzes the reversible aldol cleavage of fructose-1,6-biphosphate and fructose 1-phosphate to dihydroxyacetone phosphate and either glyceraldehyde-3-phosphate or glyceraldehyde, respectively. | -0.546658025 | -0.597643286 |
| ANXA3 | annexin A3 (ANXA3) | This gene encodes a member of the annexin family. Members of this calcium-dependent phospholipid-binding protein family play a role in the regulation of cellular growth and in signal transduction pathways. This protein functions in the inhibition of phospholipase A2 and cleavage of inositol 1,2-cyclic phosphate to form inositol 1-phosphate. This protein may also play a role in anti-coagulation. | 0.822525971 | 1.202492528 |
| B4GALT5 | beta-1,4-galactosyltransferase 5 (B4GALT5) | This gene is one of seven beta-1,4-galactosyltransferase (beta4GalT) genes. They encode type II membrane-bound glycoproteins that appear to have exclusive specificity for the donor substrate UDP-galactose; all transfer galactose in a beta1,4 linkage to similar acceptor sugars: GlcNAc, Glc, and Xyl. Each beta4GalT has a distinct function in the biosynthesis of different glycoconjugates and saccharide structures. | 0.418552469 | 0.829685649 |
| CAMP | cathelicidin antimicrobial peptide (CAMP) | This gene encodes a member of an antimicrobial peptide family, characterized by a highly conserved N-terminal signal peptide containing a cathelin domain and a structurally variable cationic antimicrobial peptide, which is produced by extracellular proteolysis from the C-terminus. The protein plays an important role in innate immunity defense against viruses. In addition to its antibacterial, antifungal, and antiviral activities, the encoded protein functions in cell chemotaxis, immune mediator induction, and inflammatory response regulation. | 0.772005697 | 0.912300975 |
| CBX7 | chromobox 7 (CBX7) | This gene encodes a protein that contains the CHROMO (CHRomatin Organization MOdifier) domain. The encoded protein is a component of the Polycomb repressive complex 1 (PRC1), and is thought to control the lifespan of several normal human cells. | -0.408857983 | -0.672227052 |
| CCNA2 | cyclin A2 (CCNA2) | The protein encoded by this gene belongs to the highly conserved cyclin family, whose members function as regulators of the cell cycle. This protein binds and activates cyclin-dependent kinase 2 and thus promotes transition through G1/S and G2/M. | 0.455750226 | 0.866349784 |
| DEFA4 | defensin alpha 4 (DEFA4) | Defensins are a family of antimicrobial and cytotoxic peptides thought to be involved in host defense. They are abundant in the granules of neutrophils and also found in the epithelia of mucosal surfaces such as those of the intestine, respiratory tract, urinary tract, and vagina. The protein encoded by this gene, defensin, alpha 4, is found in the neutrophils; it exhibits corticostatic activity and inhibits corticotropin stimulated corticosterone production. | 0.540173033 | 0.767698011 |
| EIF4B | eukaryotic translation initiation factor 4B (EIF4B) | Enables RNA binding activity. Predicted to be involved in eukaryotic translation initiation factor 4F complex assembly and formation of translation preinitiation complex. Located in cytosol. Biomarker of autism spectrum disorder and major depressive disorder. | -0.53424793 | -1.122590172 |

*(Continued)*

**Table 2.** (Continued)

| Gene | Gene name (detail) | Gene description | RA combined effect size | SLE combined effect size |
|------|--------------------|------------------|-------------------------|--------------------------|
| GPS2 | G protein pathway suppressor 2 (GPS2) | This gene encodes a protein involved in G protein-mitogen-activated protein kinase (MAPK) signaling cascades. When overexpressed in mammalian cells, this gene could potently suppress a RAS- and MAPK-mediated signal and interfere with JNK activity, suggesting that the function of this gene may be signal repression. | -0.728550581 | -0.504034151 |
| HIVEP2 | HIVEP zinc finger 2 (HIVEP2) | This gene encodes a member of a family of closely related, large, zinc finger-containing transcription factors. The encoded protein regulates transcription by binding to regulatory regions of various cellular and viral genes that maybe involved in growth, development and metastasis. | -0.553931641 | -0.408378921 |
| ID3 | inhibitor of DNA binding 3, HLH protein (ID3) | The protein encoded by this gene is a helix-loop-helix (HLH) protein that can form heterodimers with other HLH proteins. However, the encoded protein lacks a basic DNA-binding domain and therefore inhibits the DNA binding of any HLH protein with which it interacts. | -0.675277845 | -0.745779182 |
| IDH3B | isocitrate dehydrogenase (NAD (+)) 3 non-catalytic subunit beta (IDH3B) | The protein encoded by this gene is the beta subunit of one isozyme of NAD (+)-dependent isocitrate dehydrogenase. | -0.634854843 | -0.423003627 |
| IMPDH2 | inosine monophosphate dehydrogenase 2 (IMPDH2) | This gene encodes the rate-limiting enzyme in the de novo guanine nucleotide biosynthesis. It is thus involved in maintaining cellular guanine deoxy- and ribonucleotide pools needed for DNA and RNA synthesis. | -0.930897806 | -0.680928042 |
| KLHL25 | kelch like family member 25 (KLHL25) | Involved in protein ubiquitination; regulation of translational initiation; and ubiquitin-dependent protein catabolic process. | -0.547159894 | -0.553192794 |
| LDOC1 | LDOC1 regulator of NFKB signaling (LDOC1) | The gene has been proposed as a tumor suppressor gene whose protein product may have an important role in the development and/or progression of some cancers. | -0.459418035 | -0.561365132 |
| LHFPL2 | LHFPL tetraspan subfamily member 2 (LHFPL2) | This gene is a member of the lipoma HMGIC fusion partner (LHFP) gene family, which is a subset of the superfamily of tetraspan transmembrane protein encoding genes. | 0.545276617 | 0.830655827 |
| MAPRE1 | microtubule associated protein RP/EB family member 1 (MAPRE1) | The protein associates with components of the dynactin complex and the intermediate chain of cytoplasmic dynein. Because of these associations, it is thought that this protein is involved in the regulation of microtubule structures and chromosome stability. This gene is a member of the RP/EB family. | -0.476053704 | 0.40891743 |
| NFIL3 | nuclear factor, interleukin 3 regulated (NFIL3) | The protein encoded by this gene is a transcriptional regulator that binds as a homodimer to activating transcription factor (ATF) sites in many cellular and viral promoters. The encoded protein represses PER1 and PER2 expression and therefore plays a role in the regulation of circadian rhythm. | 0.523637677 | 1.073913485 |
| NMT2 | N-myristoyltransferase 2 (NMT2) | This gene encodes one of two N-myristoyltransferase proteins. N-terminal myristoylation is a lipid modification that is involved in regulating the function and localization of signaling proteins. The encoded protein catalyzes the addition of a myristoyl group to the N-terminal glycine residue of many signaling proteins, including the human immunodeficiency virus type 1 (HIV-1) proteins, Gag and Nef. | -0.627008356 | -0.772803087 |
| OASL | 2'-5'-oligoadenylate synthetase like (OASL) | Enables DNA binding activity and double-stranded RNA binding activity. Involved in several processes, including interleukin-27-mediated signaling pathway; negative regulation of viral genome replication; and positive regulation of RIG-I signaling pathway. | 0.405973339 | 1.889199094 |
| POLG2 | DNA polymerase gamma 2, accessory subunit (POLG2) | This protein enhances DNA binding and promotes processive DNA synthesis. Mutations in this gene result in autosomal dominant progressive external ophthalmoplegia with mitochondrial DNA deletions. | -0.462825481 | -0.632403304 |
| RBL2 | RB transcriptional corepressor like 2 (RBL2) | Enables promoter-specific chromatin binding activity. Involved in regulation of lipid kinase activity. Acts upstream of or within negative regulation of gene expression. | -0.820817384 | -0.664047724 |

(*Continued*)

**Table 2.** (Continued)

| Gene | Gene name (detail) | Gene description | RA combined effect size | SLE combined effect size |
|---|---|---|---|---|
| SERTAD2 | SERTA domain containing 2 (SERTAD2) | Predicted to enable transcription coactivator activity. Acts upstream of or within negative regulation of cell growth. Located in cytosol and nucleoplasm. | -0.626126151 | -0.608594971 |
| SLC22A4 | solute carrier family 22 member 4 (SLC22A4) | The encoded protein is an organic cation transporter and plasma integral membrane protein containing eleven putative transmembrane domains as well as a nucleotide-binding site motif. Transport by this protein is at least partially ATP-dependent. | 0.601572511 | 0.829446154 |
| SRGN | serglycin (SRGN) | This gene encodes a protein best known as a hematopoietic cell granule proteoglycan. This encoded protein was found to be associated with the macromolecular complex of granzymes and perforin, which may serve as a mediator of granule-mediated apoptosis. | 0.589594132 | 0.90767445 |
| TNFAIP6 | TNF alpha induced protein 6 (TNFAIP6) | This gene can be induced by proinflammatory cytokines such as tumor necrosis factor alpha and interleukin-1. Enhanced levels of this protein are found in the synovial fluid of patients with osteoarthritis and rheumatoid arthritis. | 0.723657312 | 1.336283459 |
| USP13 | ubiquitin specific peptidase 13 (USP13) | Enables several functions, including BAT3 complex binding activity; chaperone binding activity; and cysteine-type peptidase activity. Involved in several processes, including maintenance of unfolded protein involved in ERAD pathway; regulation of cellular catabolic process; and regulation of transcription, DNA-templated. Acts upstream of or within protein deubiquitination and protein stabilization. | -0.408076457 | -0.726185249 |
| VEGFB | vascular endothelial growth factor B (VEGFB) | This gene encodes a member of the PDGF (platelet-derived growth factor)/VEGF (vascular endothelial growth factor) family. Studies in mice showed that this gene was co-expressed with nuclear-encoded mitochondrial genes and the encoded protein specifically controlled endothelial uptake of fatty acids. | -0.436833473 | -0.517527546 |

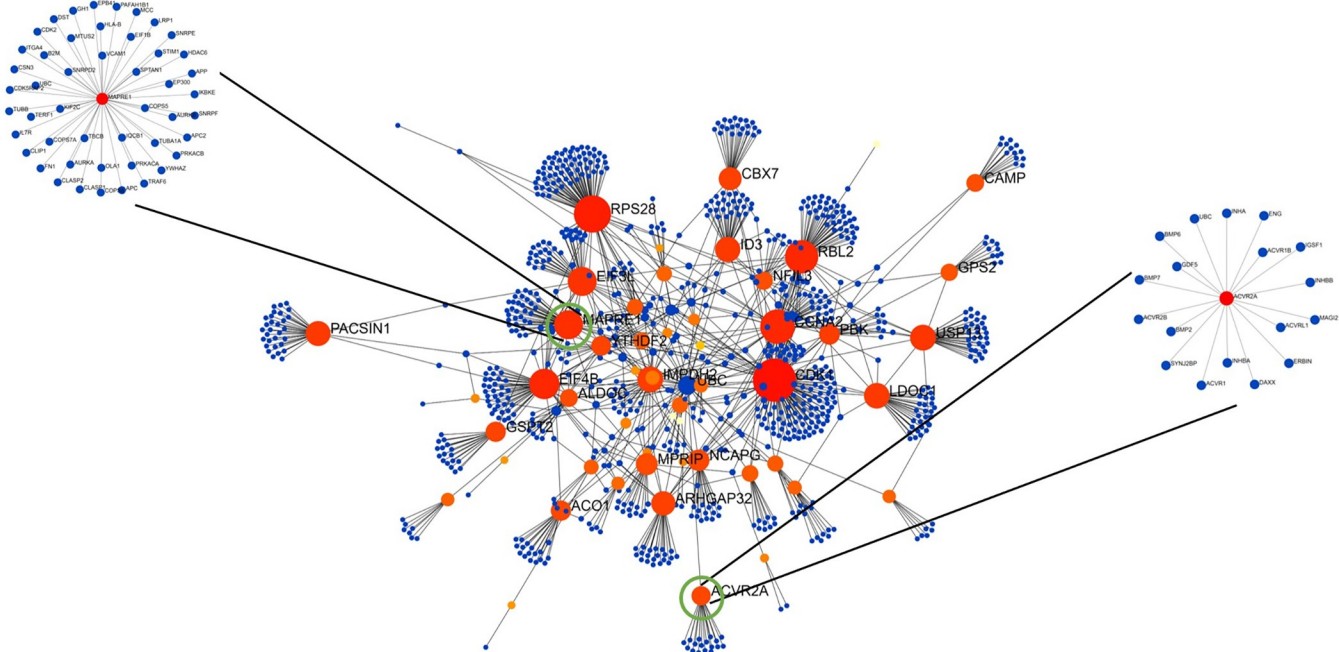

**Fig 5. Protein-Protein Interaction network of gene signatures common between RA and SLE.** The most highly ranked nodes were CDK1 (degree: 146, betweenness: 112412.8), RPS28 (degree: 94, betweenness: 66300.6), and CCNA2 (degree: 77, betweenness: 46749.4). The size and the colour of the nodes were layout by the degree and betweenness values.

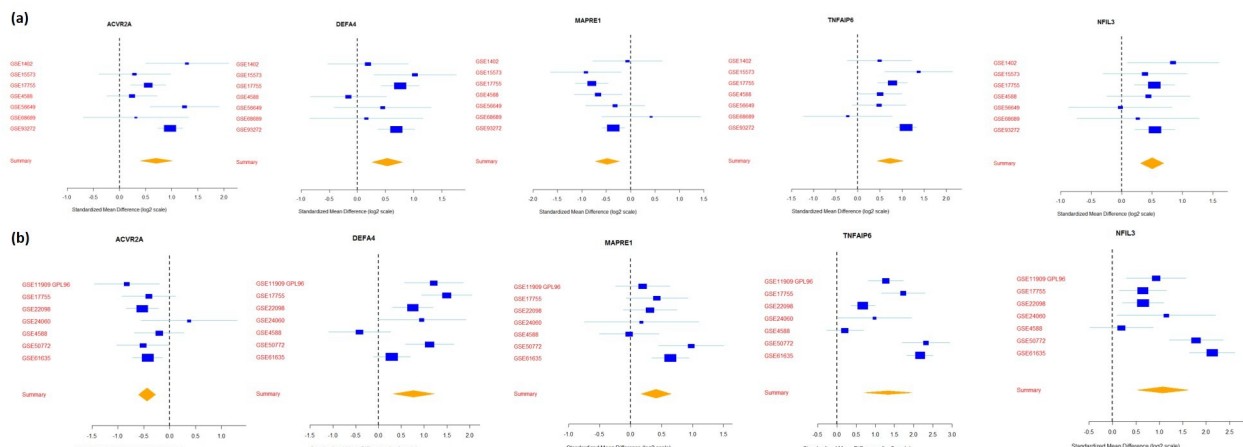

**Fig 6.** Forest plots of genes with persistent expression in all studies of RA **(a)** and SLE **(b)**. The x-axis shows the standardized mean difference (log2 scale) computed as Hedges' g between disease and control samples for genes in multiple studies. The size of the blue box is inversely proportional to the standardized mean difference of the gene in each study. Whiskers represent 95% confidence intervals. The yellow diamond represents the combined mean difference for each gene and its width denotes the 95% confidence interval.

## Discussion

The etiology and pathogenesis of RA and SLE involve different types of cells such as macrophages, T and B cells, fibroblasts, and dendritic cells, in addition to various signaling pathways and immune modulators, which make it challenging to understand the underlying mechanism

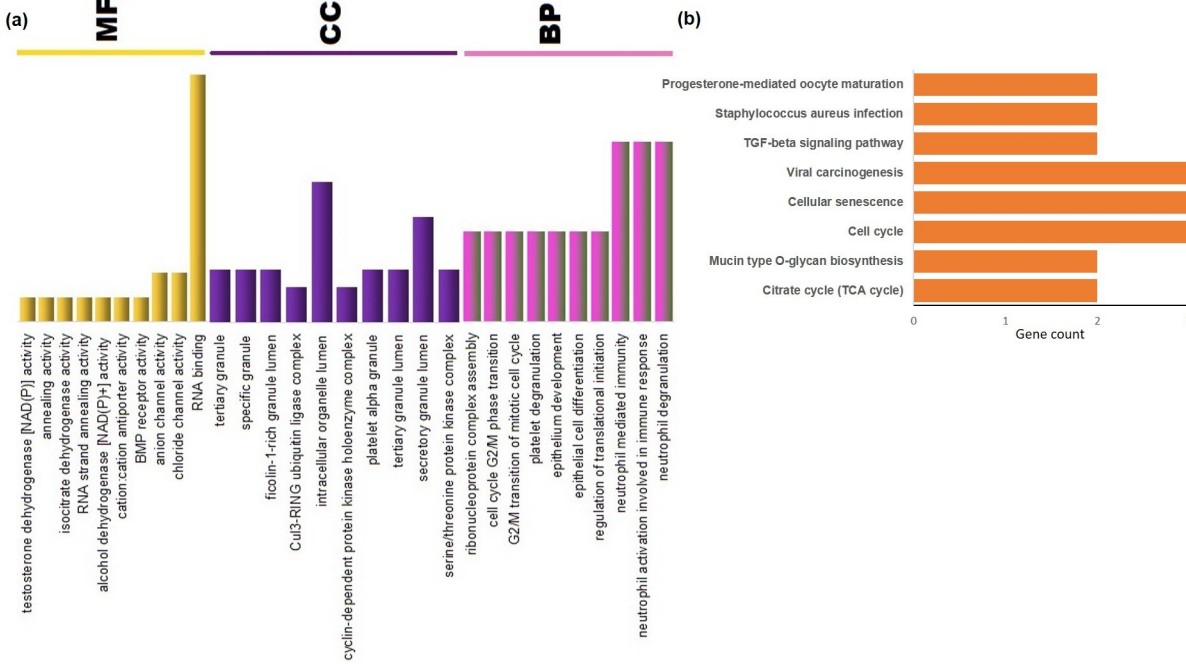

**Fig 7. Enriched GO terms and biological pathways related to common genes (P-value <0.05). (a)** The top 10 GO terms for each category (Molecular Function (MF), Cellular Component (CC), and Biological Processes (BP)) are shown. The X-axis represents the enriched GO categories and the Y-axis shows the gene counts. **(b)** The top enriched KEGG pathways. The X-axis represents the enriched KEGG pathways and the Y-axis shows the no. of genes present in the respective pathway.

for the two diseases. The present study aimed to elucidate robust common, RA and SLE-specific gene signatures by integrating gene expression data from multiple heterogeneous sources leveraging the biological (samples from different cell types) and technical heterogeneity (data generated using diverse microarray platforms). To minimize the impact caused by differences in study design and platform usage among different datasets, MetaIntegrator calculated the combined effect size by applying a random effect model. It achieves more consistent and accurate results by considering the direction and magnitude of gene expression changes.

MetaIntegrator has been successfully applied to study various diseases, from cancer to many autoimmune diseases and a few of these study outcomes have been validated in clinical settings [29–36]. Using MetaIntegrator, we analyzed 14 datasets consisting of 1,088 samples that were collected from 5 countries, 9 research centres and represented different cell types such as WB, PBMCs, and CD4 T and B immune cells to identify gene signatures which are robust and consistently differentially expressed across all studies.

This is the first study to perform a combined analysis of RA and SLE in large heterogeneous data, revealing common gene signatures systemically expressed across different cell types. Our study would find potential applications in understanding the underlying disease mechanism and exploring new biological pathways and possible drug targets for further study, which will eventually improve the understanding and management of these diseases.

The role of neutrophils in the pathogenesis of the systemic autoimmune diseases appeared as an important regulator in innate and adaptive immune responses. Neutrophils act as phagocytic cells and their role has been intensively explored in defining the pathogenesis of RA and SLE [37–43]. We identified genes *viz.* TNFAIP6 (Tumor necrosis factor-inducible gene 6 protein), ANXA3 (Annexin A3), DEFA4 (Defensin Alpha 4), and CAMP (Cathelicidin Antimicrobial Peptide) as upregulated and IMPDH2 (Inosine Monophosphate Dehydrogenase 2), ALDOC (Aldolase, Fructose-Bisphosphate C) as downregulated which are related to neutrophil-mediated immunity, activation, and degranulation. TNFAIP6, which plays a critical role in osteogenesis and bone remodeling, has previously been explored to be up-regulated in the synovial fluid of patients with rheumatoid arthritis [44]. The Defensin Alpha4 gene (DEFA4) is a member of the alpha-defensin family, a part of antimicrobial peptides in the innate immune system. Variations in DEFA4 gene expression have been reported in different disorders such as diseases related to inflammation and immunity dysfunction, brain-related disorders, and various cancers [45].

Cytokines are the main modulators of immunity. We observed YTHDF2 (YTH N6-Methyladenosine RNA Binding Protein 2) and GPS2 (G Protein Pathway Suppressor 2) in our common gene signatures negatively regulate cytokine-mediated signaling pathway, which in turn regulates the expression of Polymorphonuclear neutrophils (PMNs) and plays an important role in host defense response and inflammation. Natural Killer (NK) cells are important cells of innate immunity and their role has already been explored in the pathogenesis and etiology of various autoimmune diseases [46]. We observed that NFIL3 (Nuclear Factor, Interleukin 3 Regulated), a key immunological transcription factor that is an essential component in developing precursor NK cells, was upregulated in our study for both diseases. Interferons are a category of functionally related cytokines implicated in the pathogenesis of several rheumatic diseases. Type 1 interferon pathway has been reported to be associated with increased inflammatory response in various rheumatic conditions in response to increased expression of type 1 Interferon Stimulated Genes (ISGs) [47]. In SLE, we found many interferon related genes such as IFI27 (Interferon alpha-inducible protein 27), IFI16 (Gamma-interferon-inducible protein 16), IFI27L1 (Interferon Alpha Inducible Protein 27 like 1), IFNAR1 (Interferon alpha/beta receptor 1), IFI6 (Interferon Alpha Inducible Protein 6), IFI44 (Interferon Induced Protein 44), IFIT1-3, 5 (Interferon Induced Protein with Tetratricopeptide Repeats) which were all

upregulated. Additionally, Interferon Response Factors (IRF) such as IRF7 and IRF9 which coordinate type 1 interferon and ISGs expression were upregulated in SLE. However, in RA we observed normal expression levels for genes related to interferon. Reduced relative expression of ISGs in the circulation of RA patients as compared to SLE has already been reported [20, 48]. Even in SLE, Niewold et al. reported a wide range of serum interferon activity with 40–50% of SLE patients showing normal levels of serum interferons [49]. Therefore, status of Type 1 interferon signature as a predictive biomarker in various autoimmune conditions is debatable as it remains relatively stable in blood. The Type 1 interferon signature could play an important role in disease initiation rather than in predicting disease flares where other non-Type 1 interferon genes are reported to strongly correlate with disease activity [50].

Cell division in multicellular organisms is critical to developing and maintaining tissue homeostasis. Deregulation of cell functions leads to loss of tolerance and the development of autoimmunity [51]. Many cell cycle regulators, including cyclin-dependent kinase (CDK) and cyclins, are known for their crucial role in cell division [52]. In the common gene signatures, we identified CDK1 (Cyclin Dependent Kinase 1), CCNA2 (Cyclin A2) and MAPRE1(Microtubule Associated Protein RP/EB Family Member 1) genes that have an important role in cell division.

Bone mineralization is essential for the hardness and strength of the bone. Bone is the target tissue in inflammatory diseases, including rheumatic diseases such as RA, SLE, psoriatic arthritis and ankylosing spondylitis [53]. As bone loss has been found in both diseases, the regulation process of bone mineralization is important [54]. We found SRGN (Serglycin), known for negative regulation of bone mineralization, to be upregulated in both RA and SLE.

Ubiquitination is a key regulatory process that controls innate and adaptive immune responses. It is involved in the development, activation and differentiation of T-cells and B-cells, thus maintaining the efficient adaptive immune responses to pathogens and immunological tolerance to the self-tissues [55, 56]. In our study, we observed a negative regulator of protein polyubiquitination, GPS2, which could disrupt many aspects of immune functions and different intracellular signaling pathways.

The most striking observation was the antagonistic gene expression profiles for three genes, *i.e.* MAPRE1, ACVR2A (Activin a Receptor Type 2A), and FAM135A (Family with Sequence Similarity 135 Member A) in RA and SLE. MAPRE1 is an important gene believed to be involved in regulating microtubule structure and chromosome stability. Microtubules are important as they play an important role in maintaining cell structure [57] along with their recently identified roles in the innate and adaptive immune systems [58]. The PPI network for the common genes identified MAPRE1 as the hub gene. Hub genes produce proteins that can interact with many other proteins [59]. Hub genes play an important role in the pathogenesis and progression of many diseases; therefore, they can be targeted as diagnostic markers and candidate drug targets.

MAPRE1 was found to be upregulated in SLE whereas downregulated in RA. ACVR2A, involved in the TGF-beta signaling pathway, was also predicted to be the hub gene. TGF-beta signaling pathway plays a crucial role in immune regulation, tissue regeneration and many components of the immune system [60–63]. Malfunctioning of the TGF-beta signaling pathway can lead to immune dysregulation and other congenital effects. ACVR2A expression was elevated in RA, while it was repressed in SLE, and FAM135A followed a similar trend.

Contrary to our findings, ACVR2A expression was reported to be elevated in rheumatic diseases; however, conclusions were drawn from a small sample size of 60 patients [64]. This warrants further research into the role of these genes in the pathophysiology of autoimmune diseases. These discrepancies further emphasize the significance of using a rigorous integrated multi-cohort analysis approach. We created forest plots for ACVR2A, DEFA4, MAPRE1,

TNFAIP6, and NFIL3 genes to represent the persistent gene expression patterns across all datasets of RA and SLE. However, it is apparent that some minor deviations existed for some of the datasets, which can be further validated via inclusion of more datasets.

This study has some limitations, as it relies on publicly available datasets, thus incorporating the inherent limitations of the experimental procedures and computational methods used for data analysis. For some cell types, the sample sizes were limited, making it hard to balance the samples. The gene signature set includes too many genes to be included in a simple diagnostic test. An accurate signature based on a small set of genes would be cost-effective and more technically feasible for diagnostic purposes. The performance of the identified gene signatures in a large, prospective cohort remains unknown and requires validation on larger sample datasets further to ensure the applicability of our findings in clinical settings.

## Conclusions

With limited knowledge available about the etiology of RA and SLE, it becomes imperative to understand the precise molecular mechanisms underlying the pathophysiology of these autoimmune diseases. Many common DEGs such as TNFAIP6, DEFA4, YTHDF2, NFIL3, and SRGN predicted in our meta-analysis study have already been validated to potentially participate in the development and progression of both the diseases, which further strengthens the credibility of our results. Our study explored the novel common molecular mechanisms underlying the disease pathogenesis, and the predicted genes have the potential to be utilized as diagnostic and therapeutic targets when validated in a large prospective cohort.

## Supporting information

**S1 Table. Details of inclusion/exclusion criteria used in the study.**
(XLSX)

**S2 Table. Expression details of the 62 common gene signatures in RA and SLE.**
(XLSX)

**S3 Table. A list of 50 most significantly up or downregulated genes for RA and SLE.**
(DOCX)

**S4 Table. Detailed summary of the hub genes for the common and disease-specific gene signatures.**
(XLSX)

**S5 Table. The GO-term and enriched pathway details for common and disease-specific (RA and SLE) gene signatures.**
(XLSX)

**S1 Fig.** Precision recall Curves for RA **(a)** and SLE **(b)**. The average precision ranges from the frequency of positive examples ranging from 0.5 (for balanced data) to 1.0 (perfect model). Here, the precision-recall curves represent individual studies from PBMC, WB, and CD4 T and B cell samples.
(TIF)

**S2 Fig.** Heatmaps represent the effect size of differentially expressed gene signatures across all datasets **(a)** top RA DEGs and **(b)** top SLE DEGs. (Filtering criteria: Effect size $\geq 0.4$ and $FDR \leq 0.05$). Each column is a dataset and each row represents the expression level of the particular gene in all datasets. The colour scale represents the pooled effect size of that

particular gene ranging from yellow (low expression) to red (high expression).
(TIF)

**S3 Fig. Protein-Protein Interaction networks of DEGs for RA. (a)** Upregulated RA genes,
**(b)** Downregulated RA genes. The size and the colour of the nodes are layout by the degree
and betweenness values.
(TIF)

**S4 Fig. Protein-protein interaction networks of DEGs for SLE. (a)** Upregulated SLE genes,
and **(b)** Downregulated SLE genes. The size and the colour of the nodes are layout by the
degree and betweenness values.
(TIF)

# Acknowledgments

We acknowledge ICGEB for providing the necessary infrastructure and facilities for the
research. Senior Research Fellowship awarded to NT by GlaxoSmithKline (GSK, India), is
duly acknowledged.

# Author Contributions

**Conceptualization:** Neetu Tyagi, Dinesh Gupta.

**Data curation:** Neetu Tyagi.

**Formal analysis:** Neetu Tyagi.

**Funding acquisition:** Dinesh Gupta.

**Investigation:** Dinesh Gupta.

**Methodology:** Dinesh Gupta.

**Resources:** Dinesh Gupta.

**Supervision:** Kusum Mehla, Dinesh Gupta.

**Visualization:** Neetu Tyagi, Kusum Mehla.

**Writing – original draft:** Neetu Tyagi.

**Writing – review & editing:** Kusum Mehla, Dinesh Gupta.

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
