## [Decision Letter · Decision Letter 0]

22 Nov 2022

PONE-D-22-29923Deciphering novel common gene signatures for Rheumatoid Arthritis and Systemic Lupus Erythematosus by integrative analysis of transcriptomic profilesPLOS ONE

Dear Dr. Gupta,

Thank you for submitting your manuscript to PLOS ONE. After careful consideration, we feel that it has merit but does not fully meet PLOS ONE’s publication criteria as it currently stands. Therefore, we invite you to submit a revised version of the manuscript that addresses the points raised during the review process.

We look forward to receiving your revised manuscript.

Kind regards,

Veena Taneja

Academic Editor

PLOS ONE

Journal Requirements:

Additional Editor Comments:

Manuscript needs to provide limitations of the analysis. Data needs to account for the heterogeneity of the population, disease and the cell population used. Also, reasons for excluding any data set needs to be included.

While both RA and lupus are autoimmune diseases, the rationale of discovering common gene expression between the 2 diseases needs to be explained.

Reviewers' comments:

Reviewer's Responses to Questions

**Comments to the Author**

1. Is the manuscript technically sound, and do the data support the conclusions?

Reviewer #1: Partly

Reviewer #2: Partly

2. Has the statistical analysis been performed appropriately and rigorously? 

Reviewer #1: Yes

Reviewer #2: No

3. Have the authors made all data underlying the findings in their manuscript fully available?

Reviewer #1: Yes

Reviewer #2: Yes

4. Is the manuscript presented in an intelligible fashion and written in standard English?

Reviewer #1: Yes

Reviewer #2: Yes

5. Review Comments to the Author

Reviewer #1: In this manuscript, Tyagi et al. identify shared gene expression signatures between patients with Rheumatoid arthritis (RA) and Systemic lupus erythematosus (SLE) through a meta-analysis of publicly available fourteen microarray gene expression datasets. As mentioned in the manuscript, the main objective of identifying shared gene signatures between both autoimmune diseases is to better understand the shared mechanism of both disorders. It’s essential to identify common gene expression signatures for RA and SLE to understand the relevant biological processes that may play important roles in the shared development of these pathologies. However, in addition to identifying shared gene expression signatures for RA and SLE, identifying genes exclusive to either disease would be more critical for developing better treatment for both autoimmune diseases.

Major comment:

1. Why have only two autoimmune diseases (RA and SLE) been considered to identify the shared gene expression signatures? If the study aimed to identify the shared gene signatures in autoimmune diseases. Wouldn't it be better to consider more autoimmune diseases in the analysis?

2. As mentioned in the manuscript, a meta-analysis was performed to identify common gene expression signatures for RA and SLE. However, based on the information provided in the Methods section, I don't think the authors have performed a meta-analysis. A meta-analysis, by definition, is a statistical analysis that combines the results of multiple scientific studies addressing the same question.

3. Combining the microarray gene expression profiles of patients (RA and SLE) and healthy controls from the different studies would cause batch effects, which can lead to inaccurate conclusions.

4. In the study, why only 14 gene expression datasets were considered while many more datasets from patients with RA and SLE are available here: https://doi.org/10.1186/s12859-021-04268-4;
https://adex.genyo.es/?

5. L.155–157. Given the fact that there is a vast difference in the number of differentially expressed genes (DEGs) between RA and SLE, only 62 commonly expressed genes between both diseases do not indicate the shared biological mechanism in both autoimmune disorders.

6. Suppose someone is interested in knowing the shared expressed genes between autoimmune diseases. In that case, they could get that information from Autoimmune Diseases Explorer (https://adex.genyo.es) database. What advantage your study provides over the “Autoimmune Diseases Explorer” database?

Minor comments:

1. L.101. What is “individual meta-analysis”?

2. The quality of the figures is not good, but this could be an issue of the resolution being compromised when generating the review document.

3. Figure 2 needs to be more precise; it should be remade to see the number of commonly expressed genes between RA and SLE.

Reviewer #2: In this paper, the authors have investigated common molecular disease signatures between RA and SLE by performing a meta-analysis of publicly available microarray gene expression datasets. Overall, a multi-cohort analysis of 1088 transcriptomic profiles from 14 independent studies have been selected and used to identify common gene signatures. The authors have identified sixty-two genes common among RA and SLE, out of which fifty-nine genes had similar expression profiles in the diseases. However, opposite expression profiles were observed for ACVR2A, FAM135A, and MAPRE1 genes. Thirty genes common between RA and SLE were proposed as robust gene signatures, with persistent expression in all the studies and cell types. These gene signatures were found to be involved in innate as well as adaptive immune responses, bone development and growth.

Major issues

-The study is based upon the meta-analysis of transcriptomic datasets that have been generated on hugely different immune cells populations, i.e. whole blood, PBMCs, CD4 T cells and B cells. Meta-analyses should be performed on homogenous datasets of data, as it is highly misleading to integrate datasets that include a variety of different immune subpopulations.

It would be much more insightful to work on homogenous datasets, and also, possibly, when mixed cell populations are studied, to correct the data for the variable proportions of the different subpopulations (this could be inferred by in silico deconvolution methods)

-It is not clear why RA and SLE are being compared. The two diseases are characterized by strikingly different pathogenetic processes. What is the final aim of the study? Finding a common expression signature to be used in the diagnosis? or finding pathogenetic processes to be pharmacologically targeted? I would have rather compared more similar diseases, for instance RA, ankylosing spondylitis, psoriatic arthritis, Reiter's syndrome…

6. PLOS authors have the option to publish the peer review history of their article (what does this mean?). If published, this will include your full peer review and any attached files.

Reviewer #1: No

Reviewer #2: No

---

## [Author Response · Author response to Decision Letter 0]

6 Dec 2022

Dear Editor,

We are thankful for the insightful comments of the reviewers and for providing us an opportunity to revise our manuscript. Our response to the concerns raised by reviewers is given below.

Q.1. Manuscript needs to provide limitations of the analysis. Data needs to account for the heterogeneity of the population, disease and the cell population used. Also, reasons for excluding any data set needs to be included. While both RA and lupus are autoimmune diseases, the rationale of discovering common gene expression between the 2 diseases needs to be explained.

Ans: Thanks for the comment and suggestion. We have mentioned the limitations of our analysis in the discussion part last paragraph (page no. 23). Meta-analysis allows the integration of multiple heterogeneous datasets to identify robust and reproducible signatures by considering the heterogeneity observed in the real-world patient population. We specifically employed MetaIntegrator for the study as it shortlists robust gene signatures independent of factors such as cell type, genetic background, sex, age, etc., leveraging the biological and technical heterogeneity in these datasets. We have mentioned the exclusion criteria in the results; data pre-processing section (page no. 8). 

Autoimmune diseases share several disease mechanisms; as a result, they may have similar characteristics or etiologies. We aimed to provide shared gene signatures related to important pathological functions that can be explored for their therapeutic potential. Although many attempts were made in this direction on many autoimmune diseases, here we are focusing on the most prevalent autoimmune diseases, i.e., RA and SLE. In addition, we have conducted a meta-analysis on sufficient samples to grab the robust common gene signature that underlies the similar pathologic mechanism of the two diseases. 

Reviewer #1:

# Major comments

Q.1. Why have only two autoimmune diseases (RA and SLE) been considered to identify the shared gene expression signatures? If the study aimed to identify the shared gene signatures in autoimmune diseases. Wouldn't it be better to consider more autoimmune diseases in the analysis?

Ans: There have been several attempts to identify shared gene signatures between various autoimmune diseases, including but not limited to SLE, myositis, RA, scleroderma, Crohn's disease, ulcerative colitis, Sjogren's syndrome (SjS) and type 1 diabetes (T1D). Our study primarily focused on RA and SLE as they are highly prevalent. Even though they are heterogeneous diseases, they manifest similar clinical and pathogenic features, and thus, a differential diagnosis between these immune disorders at an early stage is not always reliable. We appreciate your idea of considering more autoimmune diseases and would certainly include them in our future studies, however it is beyond the scope of the current study.

Q.2. As mentioned in the manuscript, a meta-analysis was performed to identify common gene expression signatures for RA and SLE. However, based on the information provided in the Methods section, I don't think the authors have performed a meta-analysis. A meta-analysis, by definition, is a statistical analysis that combines the results of multiple scientific studies addressing the same question.

Ans: We have modified the methods section to clarify that the methodology adopted in this paper is indeed a meta-analysis as it combines gene expression data from various studies, all focused on identifying a set of genes to be used as disease signatures for RA and SLE. Metaintegrator achieves that by combining the effect sizes across different datasets from various studies. 

Q.3. Combining the microarray gene expression profiles of patients (RA and SLE) and healthy controls from the different studies would cause batch effects, which can lead to inaccurate conclusions.

Ans: Yes, we strongly agree with the reviewer that combining gene expression profiles from the different studies would cause batch effects and lead to an inaccurate conclusion. However, we have mentioned in the Methods section [Page no. 6, line 111; Para: Data pre-processing and meta-analysis] that MetaIntegrator deals with the problem of batch effects by estimating effect sizes for each dataset independently and then pooling these effect sizes across the datasets supplemented with Cochrane's Q value and a Fisher's p-value to account for heterogeneity.

Q.4. In the study, why only 14 gene expression datasets were considered while many more datasets from patients with RA and SLE are available here: https://doi.org/10.1186/s12859-021-04268-4;
https://adex.genyo.es/?

Ans: ADEx collects its data from the GEO only. The datasets described in ADEx were retrieved in the initial steps of our analysis but were excluded at later stages following the inclusion/exclusion criteria (Results section; Data pre-processing; page no. 8). We excluded datasets that were missing healthy controls, including drug treatment, or included gene expression from tissues such as synovial fluid, skin and other cells like monocytes, neutrophils, and other endothelial progenitor cells. The AUROC value indicates how well the MetaScore distinguished between SLE patient samples versus healthy controls. Thus datasets that decreased the summary area under the receiver operating characteristic curve (AUROC) were also excluded. 

Q.5. L.155–157. Given the fact that there is a vast difference in the number of differentially expressed genes (DEGs) between RA and SLE, only 62 commonly expressed genes between both diseases do not indicate the shared biological mechanism in both autoimmune disorders.

Ans: These 62 genes were selected based on a stringent criterion and were present across >4 studies, out of which 30 genes were significantly differentially expressed in all datasets of RA and SLE independent of age, sex, genetic background, and cell type. We focused on these genes as they hold significant importance in the disease biology of RA and SLE. We have also validated the functions of the identified genes from the literature that further ensured their important role in understanding the common pathogenesis mechanism of the two diseases.

Q.6. Suppose someone is interested in knowing the shared expressed genes between autoimmune diseases. In that case, they could get that information from Autoimmune Diseases Explorer (https://adex.genyo.es) database. What advantage your study provides over the “Autoimmune Diseases Explorer” database?

Ans: The meta-analysis framework used in ADEx uses Rank product for the identification of DEGs, which is different from the one employed in our study. In our study, we used the concept of effect size to predict a subset of genes identified as disease signatures. We also observed many novel genes in our study which were not reported earlier as common gene signatures of RA and SLE. We have used the gene expression meta-analysis tool (MetaIntegartor), which leverages biological and technical heterogeneity to identify a robust disease signature, and has been successful in diverse diseases. 

It would not be feasible to draw comparisons between the two as many of the datasets used in our study are not available in ADEx.

# Minor comments

Q.1. L.101. What is “individual meta-analysis”?

Ans: We have conducted the meta-analysis for both diseases separately by taking seven datasets of RA and 7 for SLE, termed as individual meta-analysis. To avoid further confusion, we have changed it to an independent meta-analysis.

Q.2. The quality of the figures is not good, but this could be an issue of the resolution being compromised when generating the review document.

Ans: We have rechecked the quality of figures.

Q.3. Figure 2 needs to be more precise; it should be remade to see the number of commonly expressed genes between RA and SLE.

Ans: We have generated the venn diagram to provide a more detailed view of the commonly expressed genes. It gives detailed information about the 62 genes. The figure clearly shows genes upregulated/downregulated in both diseases and genes with antagonistic expression profiles between RA and SLE. We have explored their roles in the pathogenesis of RA and SLE. 

Reviewer #2

# Major comments

Q.1. The study is based upon the meta-analysis of transcriptomic datasets that have been generated on hugely different immune cells populations, i.e. whole blood, PBMCs, CD4 T cells and B cells. Meta-analyses should be performed on homogenous datasets of data, as it is highly misleading to integrate datasets that include a variety of different immune subpopulations. 

It would be much more insightful to work on homogenous datasets, and also, possibly, when mixed cell populations are studied, to correct the data for the variable proportions of the different subpopulations (this could be inferred by in silico deconvolution methods)

Ans: The reproducibility crisis forms the crux of homogeneous studies where the findings from these studies fail when validated in multiple independent cohorts. Meta-analysis allows the integration of multiple heterogeneous datasets to identify robust and reproducible signatures by considering the heterogeneity observed in the real-world patient population. Several studies have been published by integrating data from WB and PBMCs, excluding other tissues such as synovial fluid, lung tissues etc., to ensure comparable gene expression (Toro-Domínguez D et al., 2014; Wang L et al., 2015; Luan M et al., 2017). 

We specifically used MetaIntegrator as it promises robust gene signatures independent of factors such as cell type, genetic background, sex, age, etc., by leveraging these datasets' biological and technical heterogeneity. Results of some of these studies using Metaintegrator have already been further validated in experimental settings (Khatri et al., 2013; Andres-Terre et al., 2015; Sweeney et al., 2016). 

Q.2. It is not clear why RA and SLE are being compared. The two diseases are characterized by strikingly different pathogenetic processes. What is the final aim of the study? Finding a common expression signature to be used in the diagnosis? or finding pathogenetic processes to be pharmacologically targeted? I would have rather compared more similar diseases, for instance RA, ankylosing spondylitis, psoriatic arthritis, Reiter's syndrome…

Ans: Both diseases present clinical heterogeneity, but it has been reported that both RA and SLE share common gene expression signatures in PBMC of all patients with RA and SLE (Maas et al., 2002). Thus, we aimed to compare RA and SLE to find out the genes that are common and related to their pathogenesis using a large number of datasets via a meta-analysis approach. We selected a subset of the identified genes to be robust gene signatures expressed independently of cell type, age, sex, platform, and genetic background. 

We aimed to provide shared gene signatures related to important pathologic functions that can be explored for their therapeutic potential.

References:

Toro-Domínguez D, Carmona-Sáez P, Alarcón-Riquelme ME. Shared signatures between rheumatoid arthritis, systemic lupus erythematosus and Sjögren’s syndrome uncovered through gene expression meta-analysis. Arthritis Res Ther. 2014;16: 1–8. doi:10.1186/s13075-014-0489-x

Wang L, Wu LF, Lu X, Mo XB, Tang ZX, Lei SF, et al. Integrated analyses of gene expression profiles digs out common markers for rheumatic diseases. PLoS One. 2015;10: 1–11. doi:10.1371/journal.pone.0137522

Luan M, Shang Z, Teng Y, Chen X, Zhang M, Lv H, et al. The shared and specific mechanism of four autoimmune diseases. Oncotarget. 2017;8: 108355–108374. doi:10.18632/oncotarget.19383

Khatri P, Roedder S, Kimura N, De Vusser K, Morgan AA, Gong Y, Fischbein MP, Robbins RC, Naesens M, Butte AJ, Sarwal MM. A common rejection module (CRM) for acute rejection across multiple organs identifies novel therapeutics for organ transplantation. J Exp Med. 2013 Oct 21;210(11):2205-21. doi: 10.1084/jem.20122709.

Andres-Terre M, McGuire HM, Pouliot Y, Bongen E, Sweeney TE, Tato CM, Khatri P. Integrated, Multi-cohort Analysis Identifies Conserved Transcriptional Signatures across Multiple Respiratory Viruses. Immunity. 2015 Dec 15;43(6):1199-211. doi: 10.1016/j.immuni.2015.11.003.

Sweeney TE, Wong HR, Khatri P. Robust classification of bacterial and viral infections via integrated host gene expression diagnostics. Sci Transl Med. 2016 Jul 6;8(346):346ra91. doi: 10.1126/scitranslmed.aaf7165.

Maas, K., Chan, S., Parker, J., Slater, A., Moore, J., Olsen, N. and Aune, T.M. (2002) Cutting edge: molecular portrait of human autoimmune disease. J. Immunol., 169, 5–9.

---

## [Decision Letter · Decision Letter 1]

22 Dec 2022

PONE-D-22-29923R1Deciphering novel common gene signatures for Rheumatoid Arthritis and Systemic Lupus Erythematosus by integrative analysis of transcriptomic profilesPLOS ONE

Dear Dr. Gupta,

Thank you for submitting your manuscript to PLOS ONE. After careful consideration, we feel that it has merit but does not fully meet PLOS ONE’s publication criteria as it currently stands. Therefore, we invite you to submit a revised version of the manuscript that addresses the points raised during the review process.

ACADEMIC EDITOR: The study is based on analysis of various datasets, hence it is recommended that the exclusion criteria for data not included be sound and not based on a bias outcome. The criteria for inclusion and a sound analysis strategy along with the question that is being addressed, pathogenesis vs markers of autoimmunity, needs clarification. The revision needs to address these questions diligently for acceptance. why were certain data using whole blood samples excluded and whether that decision was based on the question being addressed or other reasons?  Please define your goals clearly and provide conclusions supported strongly by the analysis. 

We look forward to receiving your revised manuscript.

Kind regards,

Veena Taneja

Academic Editor

PLOS ONE

Additional Editor Comments:

For the study to be technically sound, criteria for excluding and inclusion of various datasets needs to be detailed. How the batch effects are dealt in this analysis and the limitations of analysis needs to be explained. Whether the decision to include a dataset was made previous to analysis or excluded later poses a bias and needs to be explained.

Reviewers' comments:

Reviewer's Responses to Questions

**Comments to the Author**

1. If the authors have adequately addressed your comments raised in a previous round of review and you feel that this manuscript is now acceptable for publication, you may indicate that here to bypass the “Comments to the Author” section, enter your conflict of interest statement in the “Confidential to Editor” section, and submit your "Accept" recommendation.

Reviewer #1: (No Response)

Reviewer #2: (No Response)

2. Is the manuscript technically sound, and do the data support the conclusions?

Reviewer #1: Partly

Reviewer #2: No

3. Has the statistical analysis been performed appropriately and rigorously? 

Reviewer #1: Yes

Reviewer #2: No

4. Have the authors made all data underlying the findings in their manuscript fully available?

Reviewer #1: Yes

Reviewer #2: Yes

5. Is the manuscript presented in an intelligible fashion and written in standard English?

Reviewer #1: Yes

Reviewer #2: Yes

6. Review Comments to the Author

Reviewer #1: Major comments:

1. As per inclusion criteria, gene expression datasets from whole blood were included in the analysis. However, this needs to be clarified why the following gene expression datasets: GSE45291, GSE61635, GSE65391, GSE72509, and GSE108497, were excluded, while all of them are from whole blood. Also, please provide a supplementary table for excluded datasets mentioning the exclusion criteria based upon that dataset was excluded.

2. Authors responded that "The AUROC value indicates how well the MetaScore distinguished between SLE patient samples versus healthy controls. Thus datasets that decreased the summary area under the receiver operating characteristic curve (AUROC) were also excluded." My reply to authors' response: A decision on exclusion criteria should be made before starting the analysis, but this exclusion criteria seems like a selective exclusion of some datasets that did not work according to the authors' expectations.

Minor comments:

1. In the Table 1, GSE61635 and GSE24060 are mentioned as from PBMC and whole blood, respectively. However, these should be opposite. GSE61635 from whole blood and GSE24060 from PBMC. Please fix this typo.

Reviewer #2: The authors have not addressed any of the issues raised in the previous round of revision.

It is wrong to perform a meta-analysis of datasets generated from different mixed cell populations. Even if the method that the authors have used for the meta-analysis is based on the random model of effect size, this method can only account for small biases among datasets, such as batch effect, but it is not adeguate for integrating data from strikingly different datasets.

The authors have not even tried to better define the aims and scopes of this study, and most importantly they have not given explanation on why RA and SLE are being compared. The etiopathogenesis and response to treatment are very different for these two disorders, hence it would be much more insightful to compare more similar diseases, as previously suggested.

7. PLOS authors have the option to publish the peer review history of their article (what does this mean?). If published, this will include your full peer review and any attached files.

Reviewer #1: No

Reviewer #2: No

---

## [Author Response · Author response to Decision Letter 1]

9 Jan 2023

Dear Editor,

We are thankful for the insightful comments of the reviewers and for providing us an opportunity to revise our manuscript. Our response to the concerns raised by reviewers is given below.

Q.1. The study is based on analysis of various datasets, hence it is recommended that the exclusion criteria for data not included be sound and not based on a bias outcome. The criteria for inclusion and a sound analysis strategy along with the question that is being addressed, pathogenesis vs markers of autoimmunity, needs clarification. The revision needs to address these questions diligently for acceptance. why were certain data using whole blood samples excluded and whether that decision was based on the question being addressed or other reasons? Please define your goals clearly and provide conclusions supported strongly by the analysis. 

Ans: We have carefully tried to address all the questions raised. We have modified the manuscript to make our objective clear along with reframing the inclusion/exclusion criteria. We have also provided a supplementary table in which we have provided the exclusion details for each dataset. We have changed the conclusion section to better align with our objective.

Additional Editor Comments:

Q.2. For the study to be technically sound, criteria for excluding and inclusion of various datasets needs to be detailed. How the batch effects are dealt in this analysis and the limitations of analysis needs to be explained. Whether the decision to include a dataset was made previous to analysis or excluded later poses a bias and needs to be explained.

Ans: We have provided a supplementary table where we have given the details about the exclusion criterion for each dataset. Also, we have made changes to the manuscript for better understanding of the criteria followed. 

To maximally overcome the impact caused by the differences in study design and platform usage among different datasets or Batch effects, Combined Effect Size (ES) analysis and Random Effect Modeling (REM) were applied. This eventually results in more consistent and accurate results by taking into consideration both direction and magnitude of gene expression changes. 

We have included a paragraph in the Discussion section describing the limitations of our study. 

The decision to include/exclude a dataset has been made prior to meta-analysis, hence leaving no question of a biased approach. AUROC is used as a selection criteria to obtain a dataset that holds true discriminatory power to differentiate between healthy and diseased samples. Eventually, this dataset after meta-analysis will yield a set of genes which will serve as robust biomarkers irrespective of the heterogeneity in the real world patient population. 

Reviewers' comments:

Reviewer #1: Major comments:

1. As per inclusion criteria, gene expression datasets from whole blood were included in the analysis. However, this needs to be clarified why the following gene expression datasets: GSE45291, GSE61635, GSE65391, GSE72509, and GSE108497, were excluded, while all of them are from whole blood. Also, please provide a supplementary table for excluded datasets mentioning the exclusion criteria based upon that dataset was excluded.

Ans: We have used whole blood datasets in this study. The dataset GSE61635 is included in our study. GSE45291 dataset was excluded as this study involved patients receiving drug treatment which is one of the filtering criteria. GSE65391 was filtered based on AUROC. GSE72509 is not a part of our study as this is an RNA-Seq study and we worked with microarray datasets only. GSE108497 was excluded because it involved samples from pregnant women.

As per your suggestion, we have provided a supplementary Table S1 listing each dataset and its exclusion criterion. We have also updated the methods and results sections for the same in the manuscript.

2. Authors responded that "The AUROC value indicates how well the MetaScore distinguished between SLE patient samples versus healthy controls. Thus datasets that decreased the summary area under the receiver operating characteristic curve (AUROC) were also excluded." My reply to authors' response: A decision on exclusion criteria should be made before starting the analysis, but this exclusion criteria seems like a selective exclusion of some datasets that did not work according to the authors' expectations.

Ans: The goal of the study was to identify a set of dysregulated genes that are ideal for understanding the molecular mechanisms underlying the pathogenesis of the two most prevalent autoimmune diseases. Thus, in the first stage of filtering, we excluded datasets that involve patients with any drug treatment, unrelated/duplicated tissue samples, and missing healthy controls. The filtering at the second stage based on AUROC was crucial for our proposed objective as it ensured we were using datasets that truly discriminate between healthy vs diseased patients. Eventually, the genes predicted from this data would serve as robust biomarkers irrespective of biological/technical heterogeneity. Additionally, we have also validated the gene signatures identified in this study by taking matched AUROC filtered datasets of RA and SLE from ADEX (https://adex.genyo.es/) database. From our 62 genes list, 30 genes matched the biomarker list generated from mixed tissues ADEX datasets. This further validates the correctness of our approach. 

Minor comments:

1. In the Table 1, GSE61635 and GSE24060 are mentioned as from PBMC and whole blood, respectively. However, these should be opposite. GSE61635 from whole blood and GSE24060 from PBMC. Please fix this typo.

Ans: Thank you for pointing this out. We have corrected the Table 1 for the same.

Reviewer #2: The authors have not addressed any of the issues raised in the previous round of revision.

Q.1. It is wrong to perform a meta-analysis of datasets generated from different mixed cell populations. Even if the method that the authors have used for the meta-analysis is based on the random model of effect size, this method can only account for small biases among datasets, such as batch effect, but it is not adeguate for integrating data from strikingly different datasets.

Ans: We have designed our hypothesis based on previous studies conducted in the same direction to explore the common disease signatures for the two systemic autoimmune diseases on a large sample set. For example, Toro-Domínguez D et al., 2014, conducted a study with objective to perform a meta-analysis using publicly available gene expression data about the three diseases including rheumatoid arthritis, systemic lupus erythematosus and Sjögren’s syndrome to identify shared gene expression signatures. They have taken whole blood, PBMC and CD4 T cell and B cell datasets together for the meta-analysis to investigate the common gene expression signatures shared by the three diseases. Wang L et al., 2015, integrated data from 4 types of rheumatic diseases, and from two different tissue type. 

In our study to maximally overcome the impact caused by the differences in study design and platform usage among different datasets, Combining Effect Size (ES) analysis and Random Effect Modeling (REM) were applied to achieve more consistent and accurate results by taking into consideration both direction and magnitude of gene expression changes. Also In the process of data integration, patient samples from different sources were not segregated, for the purpose to reveal the common gene signature in the two diseases. 

We have also validated the gene signatures identified in this study by taking matched AUROC filtered datasets of RA and SLE from ADEX (https://adex.genyo.es/) database. From our 62 genes list, 30 genes matched the biomarker list generated from mixed tissues ADEX datasets. This further validates the correctness of our approach. 

Q.2. The authors have not even tried to better define the aims and scopes of this study, and most importantly they have not given explanation on why RA and SLE are being compared. The etiopathogenesis and response to treatment are very different for these two disorders, hence it would be much more insightful to compare more similar diseases, as previously suggested.

Ans: As per your suggestion, we have modified our manuscript to better define our end goal. 

We compared RA and SLE as the pathophysiology of these two is related in genetic, environmental, and immunological aspects. Previous clinical and epidemiological studies have suggested a shared genetic architecture between these two diseases. 

Additionally, several studies have been conducted in this direction in past by Daniel Toro-Domínguez et al., 2014, Icen M et al., 2009, Wang et al., 2022, Lim J 2019. A recent GWAS study revealed common genetic components between RA and SLE and provides candidate-associated loci for the understanding of the molecular mechanism underlying the comorbidity of the two diseases [Lu H et al., 2021]. These previous studies also provided evidence for the shared genetic architecture of RA and SLE. Based on these, we proposed our hypothesis to identify common gene expression patterns, hub genes, commonly regulated important pathways, and regulatory biomarkers involved in the disease mechanism of RA and SLE. 

References:

Toro-Domínguez, D., Carmona-Sáez, P. & Alarcón-Riquelme, M.E. Shared signatures between rheumatoid arthritis, systemic lupus erythematosus and Sjögren’s syndrome uncovered through gene expression meta-analysis. Arthritis Res Ther 16, 489 (2014). https://doi.org/10.1186/s13075-014-0489-x.

Icen M, Nicola PJ, Maradit-Kremers H, Crowson CS, Therneau TM, Matteson EL, Gabriel SE. Systemic lupus erythematosus features in rheumatoid arthritis and their effect on overall mortality. J Rheumatol. 2009 Jan;36(1):50-7. doi: 10.3899/jrheum.080091. PMID: 19004043; PMCID: PMC2836232.

Wang Y, Xie X, Zhang C, Su M, Gao S, Wang J, Lu C, Lin Q, Lin J, Matucci-Cerinic M, Furst DE, Zhang G. Rheumatoid arthritis, systemic lupus erythematosus and primary Sjögren's syndrome shared megakaryocyte expansion in peripheral blood. Ann Rheum Dis. 2022 Mar;81(3):379-385. doi: 10.1136/annrheumdis-2021-220066. Epub 2021 Aug 30. PMID: 34462261; PMCID: PMC8862024.

Lim J, Kim K. Genetic variants differentially associated with rheumatoid arthritis and systemic lupus erythematosus reveal the disease-specific biology. Sci Rep. 2019 Feb 25;9(1):2739. doi: 10.1038/s41598-019-39132-2. PMID: 30804378; PMCID: PMC6390106.

Lu H, Zhang J, Jiang Z, Zhang M, Wang T, Zhao H, Zeng P. Detection of Genetic Overlap Between Rheumatoid Arthritis and Systemic Lupus Erythematosus Using GWAS Summary Statistics. Front Genet. 2021 Mar 18;12:656545. doi: 10.3389/fgene.2021.656545. PMID: 33815486; PMCID: PMC8012913.

---

## [Decision Letter · Decision Letter 2]

30 Jan 2023

Deciphering novel common gene signatures for Rheumatoid Arthritis and Systemic Lupus Erythematosus by integrative analysis of transcriptomic profiles

PONE-D-22-29923R2

Dear Dr. Gupta

We’re pleased to inform you that your manuscript has been judged scientifically suitable for publication and will be formally accepted for publication once it meets all outstanding technical requirements.

Kind regards,

Veena Taneja

Academic Editor

PLOS ONE

Additional Editor Comments: 

The authors have revised the manuscript as per comments. The major concern of reviewer1 was the bias in the selection of data base. In addition, authors have included a table that details the data excluded from analysis and the exclusion criteria. Authors have included limitations of the data. Authors have included statement to explain how batch effects were dealt and revised conclusion which aligns with the objective."

Considering that the major points were revised and answered satisfactorily and  2 reviewers also think they have revised the manuscript according to suggestions, I decided to accept the second revision.

Please also find enclosed acceptance email from reviewer 1.

Reviewers' comments:

Reviewer's Responses to Questions

**Comments to the Author**

1. If the authors have adequately addressed your comments raised in a previous round of review and you feel that this manuscript is now acceptable for publication, you may indicate that here to bypass the “Comments to the Author” section, enter your conflict of interest statement in the “Confidential to Editor” section, and submit your "Accept" recommendation.

Reviewer #1: (No Response)

Reviewer #3: All comments have been addressed

2. Is the manuscript technically sound, and do the data support the conclusions?

Reviewer #1: No

Reviewer #3: Yes

3. Has the statistical analysis been performed appropriately and rigorously? 

Reviewer #1: (No Response)

Reviewer #3: Yes

4. Have the authors made all data underlying the findings in their manuscript fully available?

Reviewer #1: Yes

Reviewer #3: Yes

5. Is the manuscript presented in an intelligible fashion and written in standard English?

Reviewer #1: Yes

Reviewer #3: Yes

6. Review Comments to the Author

Reviewer #1: I don’t think that study design is appropriate, especially the dataset filtration criteria. In the second round of revisions, authors have not addressed the issues in the manuscript.

Reviewer #3: Th authors have addressed and responded to all the comments adequately from the reviewers. I have no further comments to add.

7. PLOS authors have the option to publish the peer review history of their article (what does this mean?). If published, this will include your full peer review and any attached files.

Reviewer #1: No

Reviewer #3: No

---

## [Editor Report · Acceptance letter]

6 Mar 2023

PONE-D-22-29923R2 

Deciphering novel common gene signatures for Rheumatoid Arthritis and Systemic Lupus Erythematosus by integrative analysis of transcriptomic profiles 

Dear Dr. Gupta:

I'm pleased to inform you that your manuscript has been deemed suitable for publication in PLOS ONE. Congratulations! Your manuscript is now with our production department. 

Kind regards, 

on behalf of

Dr. Veena Taneja 

Academic Editor

PLOS ONE